# Dietary Supplements Based on Red Yeast Rice—A Source of Citrinin?

**DOI:** 10.3390/toxins13070497

**Published:** 2021-07-17

**Authors:** Magdalena Twarużek, Iwona Ałtyn, Robert Kosicki

**Affiliations:** Department of Physiology and Toxicology, Faculty of Biological Sciences, Kazimierz Wielki University, 30 Chodkiewicza Street, 85-064 Bydgoszcz, Poland; twarmag@ukw.edu.pl (M.T.); robkos@ukw.edu.pl (R.K.)

**Keywords:** mycotoxin, CIT, contamination, dietary supplements, red yeast rice

## Abstract

Citrinin (CIT) is secondary metabolite of filamentous molds. This mycotoxin has nephrotoxic, hepatotoxic, embryocidal, and fetotoxic properties. It is also produced by several species of the three genera *Penicillium* spp., *Aspergillus* spp., and *Monascus* spp., which are used to make red yeast rice (RYR). The material for this study consisted of 15 dietary supplements containing an extract of fermented red rice, available on the Polish market. Samples were extracted using a MeOH–H_2_O mixture, cleaned-up with an immunoaffinity CitriTest HPLC column, and quantified by HPLC–FLD. None of the analyzed samples contained CIT above the established limit of detection (LOD). Studies on the presence of toxic metabolites in red yeast rice show the importance of regulating this product and of clear information on the label regarding the standardized amounts of monacolin.

## 1. Introduction

CIT is secondary metabolite of filamentous molds. This mycotoxin with antibiotic properties [1] was first isolated from *Penicillium citrinum* in 1931 by Hetherington and Raistrick [2]. We now know that this compound is also produced by several species of the three genera *Penicillium* spp. (*P. expansum* and *P. verrucosum*), *Aspergillus* spp. (*A. niveus* and *A. terreus*) and *Monascus* spp. (*M. ruber* and *M. purpureus*) [3,4]. *Monascus* strains are used in China to produce red and yellow food pigments [5]. Certain *Penicillium camemberti* isolates used in cheese making, and *Aspergillus oryzae* used in the production of Asian foods such as sake, miso and soy sauce, can also produce CIT [3,4]. CIT decomposes at 175 °C by dry heating, whereas with the use of a little water, the decomposition temperature decreases to 140 °C [6,7]. CIT has antibiotic properties against Gram-positive bacteria, but because of its nephrotoxic and hepatotoxic properties, it has not been used in medicine. In toxicity studies, the kidney has been identified as the major target organ of CIT after prolonged exposure [8]. Significant species differences in the susceptibility to this mycotoxin have also been observed [9,10]. The International Agency for Research on Cancer (IARC) has classified CIT as a Group 3 carcinogen due to limited evidence of its carcinogenicity in experimental animals and no evidence regarding humans. However, it increases the carcinogenicity of ochratoxin A (OTA) due to coexistence [11,12,13]. In addition to nephrotoxicity, CIT also has embryocidal and fetotoxic effects [14]. It commonly contaminates food and foodstuff, grains such as corn [15], wheat, rye, barley, oats [16] and rice [17,18]. Based on data on the presence of CIT in some supplements based on fermented rice (i.e., 15.2 mg/kg, Taiwan, and 64.7 μg/caps., the United States), the European Commission Regulation No 212/2014 amended the Regulation No 1881/2006 with regard to the maximum contamination levels of CIT in food supplements based on fermented rice *Monascus purpureus* [19,20]. Due to the nephrotoxicity and uncertainty regarding the carcinogenicity and genotoxicity of CIT, the maximum permissible content of this mycotoxin in supplements based on fermented rice was determined to be 2 mg/kg.

*Monascus purpureus* is a yeast which is used to produce red yeast rice (RYR), also known as koji, anka, angkak or ben-koji. Solid-state rice fermentation using *Monascus* is a tradition dating back to the first century AD in East Asian countries, including China, Japan and Korea [21,22]. These products, used for centuries as food and food additives, have also been applied as remedies for improving functioning of the cardiovascular system [23,24]. Additionally, in the *Chinese Pharmacopoeia*, RYR is referred to as a digestive and revitalizing remedy [25]. Red yeast rice consists of unsaturated fatty acids, phytosterols, dyes, and above all, monacolins [26], of which at least 13 have been isolated from this product. One of the monacolins present in RYR is monacolin K, which was first isolated in the second half of the 20th century by Japanese scientists. It is chemically similar to lovastatin, used to lower cholesterol [27], and is therefore believed to support lipid metabolism, lower blood pressure and have anti-inflammatory, anti-diabetic, anti-cancer and osteogenic properties [26]. In early 2014, the European Food Safety Authority (EFSA) confirmed the cause–effect relationship between the consumption of monacolin K (from fermented red rice) and maintaining proper levels of LDL cholesterol in the blood.

Randomized studies by Lu et al. [28], conducted on Chinese patients (*n* = 4870), showed that RYR reduced the risk of myocardial infarction, coronary revascularization and moderate hypercholesterolemia. The results of studies on the effects of statins on the body have shown that they cause hyperglycemia, and red rice turned out to be an alternative to these compounds. A 2014 meta-analysis including 352 patients found that RYR did not significantly increase glucose levels [27]. People who are statin-intolerant to muscle aches had lower cholesterol levels with RYR. Patients who consumed red yeast rice did not complain of gastrointestinal problems and did not show elevated levels of transaminases [23]. It has also been shown that in combination with antioxidants, RYR reduces high sensitivity to C-reactive protein (hs-CRP) and endothelial dysfunction [29], as well as effectiveness in dyslipidemia [30,31]. As a positive aspect of consuming red rice, increased osteogenic activity, cell viability and mitochondrial activity have also been reported [32]. In view of the health-promoting properties of RYR, in 2011, the EFSA decided to approve a health claim that red yeast rice products may have a pharmacotherapeutic effect [33].

## 2. Results and Discussion

Our research regarding the presence of CIT in dietary supplements containing monacolin K showed that none of the analyzed samples contained CIT exceeding the established LOD. Our results differ from those presented in previous studies. For example, a study conducted in Taiwan in 2009–2012 showed that of all 302 samples, CIT contamination was 69.0%, 35.1%, and 5.7% for raw material, dietary supplements, and processed red yeast rice products, respectively. The average contamination levels were 13.3, 1.2, and 0.1 μg/kg, respectively [34]. A similar percentage of CIT contamination was found in a study by Li et al. [35], where the percentage of contaminated samples was 28%, ranging from 16.6 to 5253 μg/kg. On the other hand, the results of Liu and Xu [36] revealed contamination at a level of 63.6%. Heber et al. [37] found this mycotoxin in seven out of nine samples; similar frequencies and levels of CIT contamination have been found in other studies [38,39,40,41]. CIT contamination of raw material and dietary supplements was 2.8–6.3 and 0.3–1.3 μg/kg, respectively [38]. In a study in Malaysia, CIT was present in 100% of RYR dietary supplement samples, with the majority of samples (76%) containing concentrations below 5 μg/kg [39]. It is worth noting the high level of CIT contamination obtained by Marley et al. [40], where five of the nine samples contained concentrations above the EU limit of 2.0 μg/kg in red fermented rice food supplements. Similarly high prevalence rates and levels of CIT were found in the analyses of 12 commercial red yeast rice products from China in the studies of Ji et al. [42]. 

## 3. Conclusions

Red yeast rice, due to its health-promoting properties, has been used for centuries as a therapeutic agent and is an increasingly used alternative in various types of therapies, including for people intolerant to statins or to lower lipids. However, the use of RYR as a dietary supplement may pose a risk of side effects due to the presence of monacolin K, which is structurally similar to lovastatin. The presence of CIT, a toxic metabolite of *Monascus* spp., can also negatively affect human health.

Although our preliminary studies did not reveal the presence of CIT above the LOD level, based on analyses by other authors and EFSA reports [43], different levels of CIT were found in RYR samples [44]. In the context of the quality and safety of dietary supplements, further research is needed to reduce the CIT content of red yeast rice. Both the FDA and the EFSA have not yet regulated all over-the-counter dietary supplements, including RYR, especially for the presence of undesirable toxic compounds. Therefore, taking this type of product should be performed with the consultation of a physician who will carefully monitor its effectiveness, safety, and tolerance.

## 4. Materials and Methods

### 4.1. Materials

The material for this study consisted of 15 dietary supplements containing extracts of fermented red rice available on the Polish market (i.e., in pharmacies, herbal shops, and online stores). The products were as follows: tablets (8), capsules with powdered content (5), capsules with oily content (1), and sachets (1). The content of monacolin K in the supplements, declared by the manufacturers, ranged from 1.5% to 4%.

### 4.2. Extraction of CIT

#### 4.2.1. Preparation of 0.01 M Phosphoric Acid

We placed 0.674 mL of 85% H_3_PO_4_ with a density d = 1.71 g/cm^3^ in a calibrated flask and added 1 dm^3^ of HPLC-grade H_2_O. With the use of a pH meter (Crison GLP 21) equipped with a pH electrode (Crison pH 5010T), the pH of the solution was adjusted to 2.5 with 2 M and 1 M NaCl solutions. Then, 750 mL of the solution was withdrawn from the flask, and the remaining volume was adjusted to pH 7.5.

#### 4.2.2. Sample Extraction

We placed 10 g of the sample in a 100 mL conical flask and homogenized it with 50 mL MeOH: H_2_O (7:3 *v/v*) for 1 min, using a Heidolph homogenizer (DIAX 900), followed by filtration through a flued filter paper. A 1 mL aliquot of the supernatant was added to 49 mL of 0.01 M H_3_PO_4_, pH = 7.5, and the mixture was filtered into a graduated cylinder using a fine fiber filter. Subsequently, 10 mL of the diluted extract (corresponding to 0.04 g of the sample) was applied to the CitriTest HPLC column at a flow rate of 1–2 drops/second. Then, the column was washed with 5 mL of 0.01 M H_3_PO_4_, pH = 7.5, and air-dried. CIT was eluted with 1 mL of MeOH: 0.01 M H_3_PO_4_ solution at pH = 2.5 (7:3 *v/v*), and the eluate was collected in a 5 mL vial; 1.5 mL of purified H_2_O was passed through the column, and the sample was vortexed.

### 4.3. Chromatographic Analysis

CIT was quantified by HPLC with fluorescence detection. The HPLC system (Merck Hitachi) consisted of an L-2130 pump, an L-2300 column oven, an L-2200 autosampler, an L-2480 fluorescence detector, and a LiChrospher^®^ 100 RP-18 column (250 × 4 mm, 5 μm) with a precolumn. The injection volume was 50 µL. The isocratic mobile phase included MeOH (55%), CH_3_COOC_2_H_5_ (10%), and 0.6 M H_3_PO_4_ (35%). The concentration was determined at a flow rate of 1 mL/min. The excitation and emission wavelengths for the fluorescence detector were 340 and 495 nm, respectively; the retention time for CIT was 4.3 min (Figure 1). The LOD and LOQ values were determined as signal-to-noise ratios of 3 and 10, respectively (Table 1). Method precision (expressed as the repeatability, percentage RSD) and recovery met the performance criteria of the analytical method in accordance with the EU Commission Regulation No. 519/2014 [45]. 

## Figures and Tables

**Figure 1 toxins-13-00497-f001:**
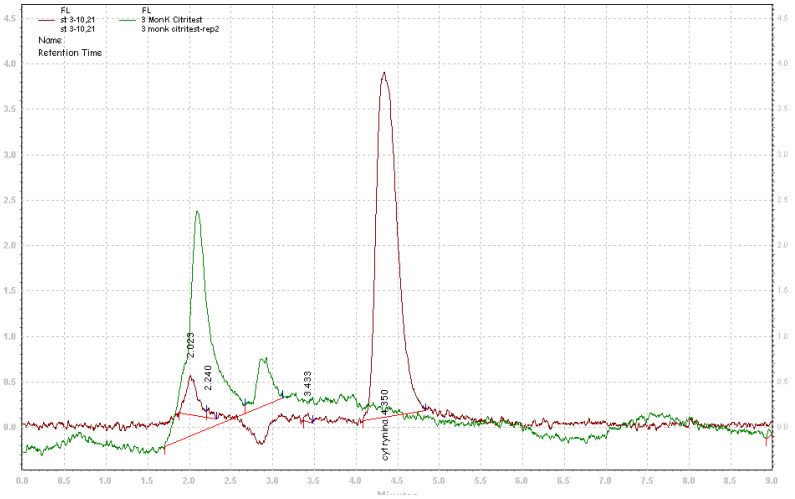
Chromatogram of CIT standard (red line) and a sample (green line).

**Table 1 toxins-13-00497-t001:** Parameters of the validation of CIT.

	Recovery (%) ± RSD (%), *n* = 3	LOD (ng/g)	LOQ (ng/g)
CIT		16.0	26.0
Spiking level—50 ng/g	89.6 ± 3.2		
Spiking level—200 ng/g	88.4 ± 3.9		
Spiking level—500 ng/g	85.3 ± 2.5		

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
