# Peer review of "Dietary Supplements Based on Red Yeast Rice—A Source of Citrinin?"

_toxins, 2021, doi:10.3390/toxins13070497_

Round 1

Reviewer 1 Report

In this article the authors evaluate the presence of citrinin in red rice supplements. Although presented as a short communication, the data presented in this article are insufficient. The authors only analyzed 15 samples, which is a very small number of samples. Logically, with this small number of samples, the authors did not find any positive samples. On the other hand, the published method does not represent any novelty over previously published methods. Therefore, I consider that the publication does not have the necessary impact for a journal such as Toxins.

Author Response

Thank you very much for reviewing the manuscript. At the time when the analysis was undertaken, only such an amount of assortment from different producers was available on the domestic market. Since CIT is a metabolite of most Penicillium molds, which also produce OTA, it is found in many raw materials and grain products, nuts and tomatoes. It was previously established (Thomas, 1931) that CIT is responsible for the "yellow rice" syndrome in Japan. It is attributed to it, like OTA, nephrotoxic and carcinogenic properties. Therefore, the authors, having only such research material, analyzed its presence.

Reviewer 2 Report

The manuscript provides information on the presence of citrinin in a specific group of foods, which are dietary supplements. I recommend that manuscript could be published after taking into account the amendments listed below.

On what basis was the method correctness for determining citrinin found? Was it the EU Commission Regulation No. 519/2014?

What values of the relative standard deviation (RSD) were estimated for the given recovery values and how were the criteria of acceptability of the result established? Recovery values are stated to two decimal places. Where does such accuracy come from? Please round to whole numbers.

If the LOD for CIT was 16 ng / g, why was the lowest enhancement was at the level of 10 ng / g?

In terms of food safety, in the context of dietary supplements, the 2017 EFSA document (doi: 10.2903 / sp.efsa.2017.EN-1177) is very important in terms of the presence of citrinin in them. I recommend that you read this document and I believe that this important document should be cited.

L49 - citation is marked in red. Please correct.

Author Response

  1. On what basis was the method correctness for determining citrinin found? Was it the EU Commission Regulation No. 519/2014? Yes, the method is compatible with EU Commission Regulation No. 519/2014. It has been added to text
  2. What values of the relative standard deviation (RSD) were estimated for the given recovery values and how were the criteria of acceptability of the result established? Recovery values are stated to two decimal places. Where does such accuracy come from? Please round to whole numbers. Values have been changed to one decimal place
  3. If the LOD for CIT was 16 ng / g, why was the lowest enhancement was at the level of 10 ng / g? This value should not be put in the text, and it is only a mistake, which was corrected and inserted the proper value
  4. In terms of food safety, in the context of dietary supplements, the 2017 EFSA document (doi: 10.2903 / sp.efsa.2017.EN-1177) is very important in terms of the presence of citrinin in them. I recommend that you read this document and I believe that this important document should be cited. It has been added to the manuscript
  5. L49 - citation is marked in red. Please correct. It has been changed

Reviewer 3 Report

Manuscript Number: toxins-1298776

Title: Dietary supplements based on Red Yeast Rice – a source of citrinin?

Type: Short Communication

This manuscript - short communication aimed to determine citrinin by HPLC-FDL in dietary supplements containing an extract of fermented red rice, available on the Polish market.

Comments:

1.

The objectives of this article were carried.

2.

The subject of the manuscript is consistent with the scope of the journal Toxins.

3.

The abstract does bring out the main points of the paper.

4.

The literature references are adequate, but It is necessary to complete next citations and discuss them:

EFSA Panel on Contaminants in the Food Chain (CONTAM); Scientific Opinion on the risks for public and  animal  health  related  to  the  presence  of  citrinin  in  food  and  feed.  EFSA Journal 2012;10(3):2605.  [82pp.] doi:10.2903/j.efsa.2012.2605. Available online: www.efsa.europa.eu/efsajournal

Mornar, A.; Serti´c, M.; Nigovi´c, B. Development of a rapid LC/DAD/FLD/MSn method for the simultaneous determination of monacolins and citrinin in red fermented rice products. J. Agric. Food Chem. 2013, 61, 1072–1080.

Ostry, V.; Malir, F.; Ruprich, J. Producers and Important Dietary Sources of Ochratoxin A and Citrinin. Toxins 2013, 5, 1574–1586.

Wang, W.; Chen, Q.; Zhang, X.; Zhang, H.; Huang, Q.; Li, D.; Yao, J. Comparison of extraction methods for analysis of citrinin in red fermented rice. Food Chem. 2014, 157, 408–412.

Silva, L.J.G.; Pereira, A.M.P.T.; Pena, A.; Lino, C.M. Citrinin in Foods and Supplements: A Review of Occurrence and Analytical Methodologies. Foods 2021, 10, 14. https://dx.doi.org/10.3390/foods10010014

Tangni, E.K.; Van Hove, F.; Huybrechts, B.; Masquelier, J.; Vandermeiren, K.; Van Hoeck, E.

Citrinin Determination in Food and Food Supplements by LC-MS/MS: Development and Use of Reference Materials in an International Collaborative Study. Toxins 2021, 13, 245. https://doi.org/10.3390/ toxins13040245

5.

The design of the study and the methodology: are appropriate. The number of analyzed samples is low (n = 15).

Why were no other validation parameters of the method given, e.g. Repeatability standard deviation (RSDr) and discussion on the availability of certified reference material?

I ask the authors for an explanation.

6.

There are minor errors:

Page 1, line 19 and and others  

The abbreviation of citrinin (CIT) is given in the text of the manuscript, it is necessary to state the abbreviation in the following text of the manuscript

Page 2, line 88  

citrinin instead citrine

I don´t have further objections. 

So, the manuscript is fit for possible publication in TOXINS journal (accept after major revision)!        

Author Response

Dear Madam/ Sir Reviewer,

here is our response and a list of changes made according to Your revision. The changes in the revised manuscript text have been identified by using the “Track Changes” function.

  1. Literature has been added
  2. The number of samples taken for analysis is low, because only such a number of supplements from various manufacturers were available on the Polish market
  3. The validation parameters had been added. Certified reference material was not available
  4. Page 1, line 19 and and others 

    The abbreviation of citrinin (CIT) is given in the text of the manuscript, it is necessary to state the abbreviation in the following text of the manuscript. Is has been changed
  5. Page 2, line 88 

    citrinin instead citrine Is has been changed 

Round 2

Reviewer 1 Report

The authors have corrected the manuscript.

Reviewer 3 Report

Manuscript Number: toxins-1298776R1

Title: Dietary supplements based on Red Yeast Rice – a source of citrinin?

Type: Short Communication

Comments

There are errors:

1.

Page 1, line 23  

Penicillium viridicatum should be replaced by Penicillium verrucosum

2.

Page 1, abstract line 10

Citrinin should be replaced by CIT

3.

Page 1, line 28, 29, 34, 40, 43

Citrinin should be replaced by CIT

4.

Page 2, line 45, 79, 80, 82, 89, 92, 94, 96

Citrinin should be replaced by CIT

5.

Page 3, line 104, 106, 108, 109, 136, 140

Citrinin should be replaced by CIT

6.

Page 3, line 104, 106, 108, 109, 136, 140

Citrinin should be replaced by CIT

7.

Page 4, Table 1

Citrinin should be replaced by CIT

8.

Page 5, line 185

Correct typos in the reference: Ostry, V.; Malir, F.; Ruprich, J. Producres and important dietary sources od ochratoxin A and citrinin. Toxins 2013, 5, 1574–1586.

So, the manuscript is fit for possible publication in TOXINS journal (accept after minor revision)!